# Comparison of the Oncological and Functional Outcomes of Brachytherapy and Radical Prostatectomy for Localized Prostate Cancer

**DOI:** 10.3390/medicina58101387

**Published:** 2022-10-02

**Authors:** Fei Wang, Yang Luan, Yaqin Fan, Tianbao Huang, Liangyong Zhu, Shengming Lu, Huazhi Tao, Tao Sheng, Deqing Chen, Xuefei Ding

**Affiliations:** 1Department of Urology, Municipal Key-Innovative Discipline, The Second Affiliated Hospital of Jiaxing University, Jiaxing 314001, China; 2Department of Urology, Northern Jiangsu People’s Hospital, Yangzhou 225001, China; 3Departments of Oncology, The First Affiliated Hospital of Jiaxing University, Jiaxing 314001, China; 4Department of Urology, Jiaxing Hospital of Traditional Chinse Medicine, Jiaxing University, Jiaxing 314001, China; 5Forensic and Pathology Laboratory, Provincial Key Laboratory of Medical Electronics and Digital Health, Institute of Forensic Science, Jiaxing University, Jiaxing 314001, China

**Keywords:** prostate cancer, prostatectomy, brachytherapy, efficacy, health-related quality of life

## Abstract

*Background and Objectives*: To compare the oncological and functional outcomes of brachytherapy (BT) and radical prostatectomy (RP) in patients with localized prostate cancer (PCa). *Materials and Methods*: We retrospectively analyzed data from 557 patients with localized PCa who were treated with BT (*n* = 245) or RP (*n* = 312) at Northern Jiangsu People’s Hospital between January 2012 and December 2017. Biochemical relapse-free survival (bRFS) and cancer-specific survival (CSS) were compared by treatment modality. Multivariate Cox regression analysis was used to evaluate bRFS. Health-related quality of life (HRQoL) was measured using the Expanded Prostate Cancer Index Composite (EPIC) questionnaire. *Results*: The BT group was older and had a higher initial PSA (iPSA). The 5-year bRFS was 82.9% in the BT group versus 80.1% in the RP group (*p* = 0.570). The 5-year CSS was 96.4% in the BT group versus 96.8% in the RP group (*p* = 0.967). Based on multivariate Cox regression analysis, Gleason score ≥ 8 was the main independent prognostic factor for bRFS. Regarding the HRQoL, compared with the baseline, both treatments produced a significant decrease in different aspects of HRQoL at 3, 6, and 12 months after treatment. Patients in the BT group had lower HRQoL with regard to urinary irritation/obstruction and bowel function or bother, while patients in the RP group had lower HRQoL concerning urinary incontinence and sexual function or bother. There was no significant difference in HRQoL aspects between the two groups after follow-up for 2 years compared with the baseline. *Conclusions*: BT provides equivalent oncological control outcomes in terms of bRFS and CSS for patients with localized PCa compared with RP. Gleason score ≥ 8 was the main independent prognostic factor for bRFS. BT had better HRQoL compared with RP, except for urinary irritation/obstruction and bowel function or bother, but returned to baseline after 2 years.

## 1. Introduction

Prostate cancer (PCa) is the highest incidence male malignancy in Western countries [1]. In recent years, as the population ages, the Westernized lifestyle is increasingly adopted, and prostate-specific antigen (PSA) screening and improved biopsy techniques are implemented, the incidence of PCa has been increasing every year in China [2]. Many treatment options can be used for localized PCa, including active surveillance (AS), external beam radiotherapy (EBRT), radical prostatectomy (RP), and brachytherapy (BT). The optimal treatment for localized PCa is still a controversial subject.

Although RP is considered a standard treatment method for localized PCa [3], poor erectile function outcomes and elevated incontinence rates represent major disadvantages [4,5]. Furthermore, aged patients and those with underlying diseases may have difficulty tolerating radical surgery.

The American Brachytherapy Society consensus guidelines suggest that BT is a safe and effective treatment for patients with localized PCa [6]. Data indicate that BT is the best choice for patients over 75 years of age [7,8]. Moreover, patients tend to place equal emphasis on the expected oncological and functional outcomes associated with each treatment modality. However, few comparative studies have examined the oncological and functional outcomes of BT and RP for localized PCa. Therefore, we conducted a single-institutional, retrospective, and comparative study evaluating oncological and functional outcomes of BT and RP for localized PCa during the same time period.

## 2. Materials and Methods

### 2.1. Patients

We evaluated 557 patients with localized PCa (T1c-T3aN0M0) who underwent BT (*n* = 245) or RP (*n* = 312) at Northern Jiangsu People’s Hospital between January 2012 and December 2017. The inclusion criteria were the following: A clinical T stage between T1c and T3a, ≥2 years follow-up posttreatment. Patients who received adjuvant radiation therapy/chemotherapy and/or patients with distant metastasis were excluded from the present study. Patients were divided into low, intermediate, and high risk according to the National Comprehensive Cancer Network (NCCN) guidelines: PSA ≤ 10 ng/mL, Gleason score ≤ 6, and stage ≤ T2a for low-risk patients, PSA 10–20 ng/mL, Gleason score 7, and stage T2b for intermediate-risk patients, and PSA > 20 ng/mL, Gleason score ≥ 8, or stage ≥ T2c for high-risk patients [9].

Patient evaluation included medical history, physical examination, initial PSA (iPSA), and transrectal ultrasound-guided biopsy. Clinical staging was based on Gleason score, digital rectal examination (DRE), iPSA, and imaging studies (bone scan, pelvic computed tomography, or magnetic resonance imaging). The therapeutic decisions were made by the surgeon according to the patient’s discussion and preference.

### 2.2. Radical Prostatectomy

After infraumbilical incision and access to the extraperitoneal space, dissection of the pelvic lymph nodes was carried out. Following prostate exposure, the endopelvic fascia was opened, with ligation and sectioning of the penis dorsal venous complex. The next step involved dissection and section of the urethra. The prostate was then dissected retrogradely, preserving the neurovascular bundle or not according to the clinical and surgical staging. Finally, Denonvillier fascia separation was performed with prostate removal and hemostasis. Vesicourethral anastomosis was performed employing a urethral catheter, which remained for 10 to 12 days.

### 2.3. Brachytherapy

Preplanning for BT was performed using the prostate volume obtained by transrectal ultrasound (Flex focus 1202; BK, Naerum, Denmark) to determine the overall activity of the radioisotope. Patients underwent epidural anesthesia in the bladder lithotomy position with an indwelling catheter before BT. The radioisotope used in all patients was iodine-125. Iodine-125 seeds were accurately introduced in preplanned positions using a brachytherapy stepping unit (Mick Radio-Nuclear Instruments, Mount Vernon, NY, USA) with a standard 0.5 cm brachytherapy template implanted via a transperineal approach. The prescribed dose was 145 Gy. Iodine-125 seeds were placed through the needles with a Mick applicator under real-time transrectal ultrasonography guidance. A plain film of the kidney–ureter–bladder was scheduled to confirm the distribution of the implanted seeds after the procedure. The urinary catheter was withdrawn 2 to 5 days after BT. Dosimetric analysis was evaluated by computed tomography (CT) for 4 weeks after implantation. A monotherapy approach with BT was used for low-risk patients; androgen deprivation therapy (ADT) was administered for intermediate-risk (4–6 months) and high-risk patients (2–3 years).

### 2.4. Follow-Up

We analyzed the follow-up data obtained by telephone follow-up survey and periodic outpatient reexamination. Follow-up visits consisting of serum PSA and DRE were scheduled every 3 months for the first year, every 6 months in the second year, and then annually thereafter. The primary endpoints to determine the oncological outcomes were biochemical relapse-free survival (bRFS) and cancer-specific survival (CSS). Biochemical recurrence for patients undergoing BT was defined as a nadir PSA + 2 ng/mL or more using the Phoenix definition (nadir + 2 ng/mL) and for those undergoing RP as two consecutive PSA ≥ 0.2 ng/mL [10,11]. The bRFS was defined as the time from the treatment to PSA recurrence or death from any cause. CSS was defined as death due to PCa or the presence of uncontrolled metastatic disease at the time of death.

Functional outcomes refer to health-related quality of life (HRQoL). HRQoL was measured in patients treated for localized PCa with RP and BT using the Expanded Prostate Cancer Index Composite (EPIC) questionnaire at baseline and 3, 6, 12, and 24 months after the treatment [12].

### 2.5. Statistical Analysis

Data were expressed as percentage or mean scores ± standard deviation. Differences between categorical variables were compared using the chi-squared test, and differences between continuous variables were compared using a *t*-test. We used the Kaplan–Meier method and the log-rank test to estimate bRFS and CSS. A Cox regression model was used for multivariate analysis of bRFS. *p* < 0.05 was considered to be statistically significant. All statistical analyses were performed using SPSS Statistics version 23.0 (IBM Corporation, Armonk, NY, USA).

## 3. Results

We initially identified 562 patients who met the inclusion criteria for the study, and five of these were subsequently excluded due to lack of follow-up. The clinical characteristics of the study population are shown in Table 1. The BT group was older (74.16 vs. 63.87 years, respectively) and had higher initial PSA (iPSA) (17.81 vs. 15.34 ng/mL), compared to the RP group. There was no statistical difference between the two groups regarding biopsy Gleason score, NCCN risk category, and clinical T stage. The mean follow-up time was 52.58 ± 20.59 months (range 3–103 months).

Biochemical recurrence occurred in 36 and 51 patients in the BT and RP groups at the time of the last follow-up visit, respectively. Eleven patients in the RP group died, nine due to PCa and two due to cerebrovascular disease. In the BT group, 16 patients died; six due to PCa, three due to digestive tract cancer, two due to cerebrovascular disease, and the others due to unknown causes.

With regard to the oncological outcomes, the 5-year bRFS was 82.9% in the BT group versus 80.1% in the RP group (*p* = 0.570; Figure 1a). When stratified according to risk, for the BT group, the 5-year bRFS for patients presenting with low-, intermediate-, and high-risk disease was 95.2%, 91.0%, and 76.1%, respectively, compared with 90.1%, 84.7%, and 74.3% in the RP group (*p* = 0.340, 0.477, 0.840, respectively; Figure 1b–d). Therefore, there was similar biochemical control in the RP and BT groups at 5 years. The 5-year CSS was 96.4% in the BT group versus 96.8% in the RP group, there was no statistically significant difference between two groups (*p* = 0.967; Figure 2).

Based on multivariate Cox regression analysis, Gleason score ≥ 8 (HR 3.669, 95% CI 2.06–6.53; *p* < 0.001) was the main independent prognostic factor for bRFS (Table 2). Treatment modality (BT vs. RP), age (>65 vs. ≤65), PSA (>10 vs. ≤10), and clinical T stage (≥T2b vs. ≤T2a) were not prognostic factors of bRFS.

The HRQoL of the two groups of patients was affected to varying degrees after treatment (Table 3). Compared with the baseline, both treatments produced a significant decrease in different aspects of HRQoL at 3, 6, and 12 months after treatment: patients in the BT group had lower HRQoL with regard to urinary irritation/obstruction and bowel function or bother, while patients in the RP group had lower HRQoL concerning urinary incontinence and sexual function or bother. There was no statistically significant difference in HRQoL aspects between the two groups after 2 years of follow-up compared with the baseline.

## 4. Discussion

The treatments of localized PCa include AS, EBRT, RP, and BT [13]. RP is considered a standard treatment for early stage PCa [3]. Due to the complete resection of the tumor and detailed pathological analysis, the surgery is selected more commonly by patients. Major advantages of RP include precise assessment of the extent of the disease at a low morbidity cost, high level of confidence in the long-term eradication, ease of detection of recurrence with a tumor marker, and availability of treatment of the long-term complications (i.e., urinary incontinence and erectile dysfunction) that affect the quality of life. Unfortunately, poor erectile function outcomes and elevated incontinence rates represent major disadvantages [4].

With the development and application of a computerized treatment planning system and new radionuclide, BT for PCa has developed rapidly. BT is a technology by which a radioactive isotope is placed inside or around the tumor. The tumor receives a high dose of radiation without elevating the dose to surrounding normal tissues. Some advantages of BT include being minimally invasive and having a definite effect and fewer complications, which may contribute to its popularity in Western countries [14]. The American Brachytherapy Society consensus guidelines suggest that BT is a safe and effective treatment for patients with localized PCa [6]. Furthermore, BT is considered a great therapeutic option for aged patients and those with complicated medical diseases who may have difficulty tolerating radical surgery [7,8].

In the present study, we analyzed 557 patients with localized PCa who underwent BT (*n* = 245) or RP (*n* = 312). The BT group was older and had a higher iPSA level. The results indicated that the 5-year bRFS rate was 82.9% (low risk: 95.2%, intermediate risk: 91.0%, and high risk: 76.1%) in the BT group versus 80.1% (low risk: 90.1%, intermediate risk: 84.7%, and high risk: 74.3%) in the RP group. Although the 5-year bRFS for RP was lower compared with BT, there was no statistically significant difference between the two groups (all *p* > 0.05). In addition, there was no significant difference between RP and BT with regard to bRFS by multivariate analysis. The 5-year CSS was 96.4% in the BT group versus 96.8% in the RP group, a non-statistically significant difference (*p* > 0.05).

Giberti et al. reported similar 5-year biochemical disease-free survival rates for RP (91.0%) or BT (91.7%) in patients with low-risk PCa [15]. Fisher et al. reported a comparative study of men with low- to intermediate-risk PCa treated with BT and RP [16]. After RP, the 5-year bRFS values were 96.1% and 90.6% for low- and intermediate-risk patients, respectively. After BT, the 5-year bRFS values were 92.5% and 95.8% for low- and intermediate-risk disease, respectively. The 5-year CSS for patients was 100% for both RP and BT. This finding argued that excellent disease control outcomes can be achieved after RP and BT for men with early stage localized PCa. Similarly, Zhang et al. indicated that BT was associated with a similar risk of biochemical recurrence rate and prostate cancer-specific mortality compared with RP for localized PCa [17]. Guo et al. reported similar 10-year cancer-specific mortality rates for RP (1.2%) and BT (2.0%) in low- to intermediate-risk PCa patients aged ≥ 70 years [18]. Guo et al. deemed that BT offers oncological outcomes similar to RP in elderly patients with localized PCa. In a recent study, Suárez et al. reported a comparative study of mortality and biochemical recurrence after RP, BT, or external radiotherapy for localized PCa patients at 10 years of follow-up, and BT presented high overall survival similarly to RP, but higher risk of biochemical progression [19]. These results were similar to our study.

There are a large number of prognostic factors of PCa, such as age, initial PSA, Gleason score, and T stage [20]. Ciezki et al. reported that clinical stage T3, biopsy Gleason score 8–10, higher pretreatment PSA, shorter ADT duration, and more frequent PSA testing following therapy were associated with significantly worse bRFS [21]. Zhou et al. reported that clinical stage ≥ T2b was associated with significantly worse bRFS [22]. Similarly, in the multivariate analysis of the present study, we also considered Gleason score ≥ 8 as the main independent prognostic factor for bRFS. The treatment modality, age, iPSA, and clinical T stage exerted no influence on bRFS.

It is necessary to consider not only cancer control but also HRQoL for patients facing the decision of which treatment to choose for localized PCa. HRQoL was measured in patients treated for localized PCa with RP and BT using the EPIC questionnaire at baseline and 3, 6, 12, and 24 months after the treatment. The EPIC is a 50-item questionnaire with eight domains, including urinary function, urinary irritation/obstruction, urinary incontinence, urinary bother, bowel function, bowel bother, sexual function, and sexual bother [12]. Each domain is scored from 0 to 100, with higher scores indicating better HRQoL. For HRQoL in this study, compared with baseline, both treatments produced a significant decrease in HRQoL in different aspects at 3, 6 months and 1 year after treatment. Patients in the BT group had lower HRQoL with regard to urinary irritation/obstruction and bowel function or bother, while patients in the RP group had lower HRQoL regarding urinary incontinence and sexual function or bother. The scores reached a nadir 3 months after treatment and then recovered. There was no significant difference in HRQoL aspects between the two groups after 2 years of follow-up.

Chen et al. reported a comparative study about the quality of life after RP, EBRT, and BT vs. AS [23]. Compared with AS, sexual dysfunction worsened by 3 months in patients who underwent RP, EBRT, and BT. Compared with AS at 3 months, worsened urinary incontinence was associated with RP, acute worsening of urinary obstruction and irritation with EBRT and BT, and worsened bowel symptoms with EBRT. By 24 months, the mean scores between the treatment groups vs. AS were not significantly different in most domains. Giberti et al. reported the functional outcomes after radical retropubic prostatectomy (RRP) versus BT for the treatment of low-risk PCa during a 5-year assessment [15]. At 6 months and 1 year, both treatments produced a significant decrease in aspects of the quality of life, while in BT patients, there was a significantly higher and longer lasting rate of urinary irritation disorders but better erectile function than in the RRP group. No differences in the functional outcomes were encountered after 5 years in either group. De et al. reported a comparative study about patient-reported outcomes over 5 years following RP and external beam radiation therapy with low-dose-rate brachytherapy boost (EBRT-LDR) for localized PCa [24]. Compared to RP, EBRT-LDR was associated with meaningfully worse urinary irritative/obstructive and bowel functions but better urinary incontinence function up to 5 years after treatment.

The incidence of urinary irritation or obstruction was higher after BT, which is related to the dose and distribution of radioactive seeds [25]. Urethral irradiation dose should be reduced as much as possible in order to reduce postoperative urinary irritation or obstruction. Furthermore, Elshaikh et al. found that prophylactic tamsulosin before BT significantly improved lower urinary tract symptoms [26]. Transurethral resection of the prostate (TURP) may be considered for recurrent urinary retention due to bladder outlet obstruction. In this study, three patients eventually required TURP because of prolonged urinary retention. BT could achieve superior genitourinary function [27], and the urinary incontinence and sexual function in the BT group was better than that in the RP group. This is because BT preserves the prostate’s anatomical structure and does not directly damage the neurovascular bundle. Therefore, BT is a potential alternative therapeutic modality to RP for patients (especially for aged patients or those with complicated medical diseases) seeking a potentially curative treatment.

This study has some limitations. First, this study is a single-institutional, retrospective study. The small number of patients evaluated and the short follow-up period may have influenced the oncological results and posttreatment HRQoL. A longer observational period is required for a meaningful comparison of overall survival time. Second, the results of this study may have created some bias due to the lack of propensity score matching analysis. Prospective, randomized studies with a larger number of patients and a longer follow-up period are required to confirm these encouraging results.

## 5. Conclusions

BT provides equivalent oncological control outcomes in terms of bRFS and CSS for patients with localized PCa compared with RP. Gleason score ≥ 8 was the main independent prognostic factor for bRFS. The BT group had better HRQoL compared with the RP group, except for urinary irritation/obstruction and bowel function or bother, with a return to baseline after 2 years. BT is a potential alternative approach to RP for patients (especially for aged patients or those with complicated medical diseases) seeking potentially curative treatment. These results could provide important information for clinical decision making for patients with PCa.

## Figures and Tables

**Figure 1 medicina-58-01387-f001:**
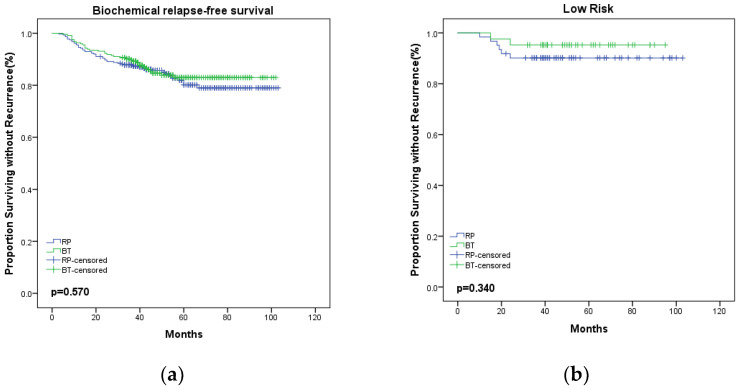
Kaplan–Meier curve for bRFS between BT and RP. (**a**) Overall, (**b**) low risk, (**c**) intermediate risk, (**d**) high risk. BT, brachytherapy; RP, radical prostatectomy; bRFS, biochemical relapse-free survival.

**Figure 2 medicina-58-01387-f002:**
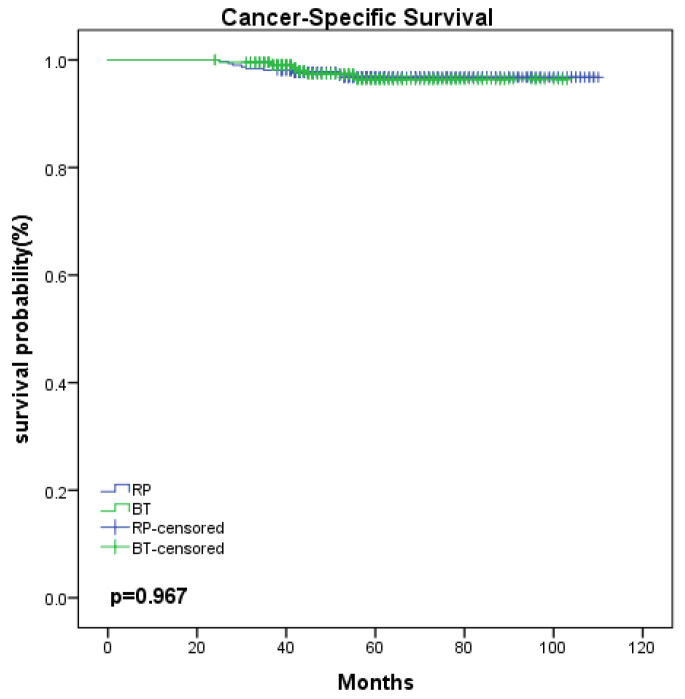
Kaplan–Meier curve for CSS between BT and RP. BT, brachytherapy; RP, radical prostatectomy; CSS, cancer-specific survival.

**Table 1 medicina-58-01387-t001:** Clinical characteristics of patients undergoing RP and BT.

Characteristic	BT (*n* = 245)	RP (*n* = 312)	Total (*n* = 557)	*p* Value
Age (years)				<0.01
Range	53–86	49–83	49–86	
Median	75	64	68	
Mean (SD)	74.16 (6.40)	63.87 (7.66)	68.40 (8.77)	
Initial PSA (ng/mL)				0.141
≤4, *n* (%)	8 (3.3)	8 (2.6)	16 (2.9)	
4–10, *n* (%)	68 (27.7)	111 (35.6)	179 (32.1)	
>10, *n* (%)	169 (69.0)	193 (61.8)	362 (65.0)	
Mean (SD)	17.81 (9.12)	15.34 (8.6)	16.43 (8.91)	<0.01
Biopsy Gleason score				0.427
≤6, *n* (%)	127 (51.8)	178 (57.0)	305 (54.8)	
7, *n* (%)	69 (28.2)	82 (26.3)	151 (27.1)	
≥8, *n* (%)	49 (20)	52 (16.7)	101 (18.1)	
Clinical T stage				0.938
T1c, *n* (%)	40 (16.3)	48 (15.4)	88 (15.8)	
T2a, *n* (%)	65 (26.5)	84 (26.8)	149 (26.8)	
T2b, *n* (%)	53 (21.6)	63 (20.2)	116 (20.8)	
T2c, *n* (%)	78 (31.9)	108 (34.6)	186 (33.4)	
T3a, *n* (%)	9 (3.7)	9 (2.9)	18 (3.2)	
NCCN risk category				0.606
Low, *n* (%)	42 (17.1)	61 (19.6)	103 (18.5)	
Intermediate, *n* (%)	61 (24.9)	83 (26.6)	144 (25.8)	
High, *n* (%)	142 (58)	168 (53.8)	310 (55.7)	

**Note:** RP, radical prostatectomy; BT, brachytherapy; PSA, prostate-specific antigen; NCCN, National Comprehensive Cancer Network; SD, standard deviation.

**Table 2 medicina-58-01387-t002:** Multivariable analyses for biochemical relapse-free survival.

Factor	Multivariate		
	HR	95% CI	*p* Value
Treatment modality			0.082
RP vs. BT	1.534	0.95–2.48	
Age (years)			0.271
>65 vs. ≤65	1.329	0.80–2.21	
iPSA (ng/mL)			0.841
>10 vs. ≤10	1.059	0.61–1.85	
Gleason score			
≤6	1	Ref.	-
7	1.574	0.91–2.72	0.105
≥8	3.669	2.06–6.53	<0.001
Clinical T stage			
≥T2b vs. ≤T2a	1.264	0.79–2.03	0.335

Note: Ref., reference; RP, radical prostatectomy; BT, brachytherapy; iPSA, initial prostate-specific antigen; CI, confidence interval; HR, hazard ratio.

**Table 3 medicina-58-01387-t003:** The EPIC scores of patients undergoing RP and BT.

	BT *n* = 245 (Mean ± SD)	RP *n* = 312 (Mean ± SD)	*p* Value (BT vs. RP)
Urinary function			
Baseline	96.4 ± 11.2	94.5 ± 10.4	0.039
3-month	87.5 ± 14.3	81.7 ± 17.6	<0.001
6-month	92.6 ± 8.9	86.3 ± 14.2	<0.001
12-month	93.9 ± 12.2	89.1 ± 13.4	<0.001
24-month	95.3 ± 10.7	93.8 ± 11.3	0.112
Urinary irritative/obstructive			
Baseline	95.1 ± 12.9	93.3 ± 11.1	0.078
3-month	81.3 ± 15.7	85.4 ± 13.6	<0.001
6-month	85.7 ± 12.5	91.4 ± 5.9	<0.001
12-month	88.3 ± 10.2	92.6 ± 4.5	<0.001
24-month	95.5 ± 8.7	94.2 ± 5.3	0.03
Urinary incontinence			
Baseline	97.9 ± 6.3	96.5 ± 7.6	0.021
3-month	94.3 ± 7.5	68.5 ± 23.5	<0.001
6-month	96.5 ± 9.7	76.9 ± 19.3	<0.001
12-month	97.5 ± 8.6	84.2 ± 16.4	<0.001
24-month	96.7 ± 8.9	95.2 ± 12.9	0.121
Urinary bother			
Baseline	94.6 ± 8.8	93.4 ± 12.1	0.193
3-month	85.3 ± 16.1	87.6 ± 15.4	0.087
6-month	88.4 ± 11.4	90.4 ± 10.2	0.03
12-month	92.6 ± 10.1	92.9 ± 9.1	0.713
24-month	93.9 ± 9.2	93.1 ± 8.7	0.294
Bowel function			
Baseline	97.5 ± 5.2	96.1 ± 7.3	0.011
3-month	95.1 ± 8.5	95.2 ± 8.8	0.893
6-month	95.9 ± 8.7	94.9 ± 7.5	0.146
12-month	96.3 ± 6.1	95.3 ± 6.7	0.070
24-month	97.0 ± 5.7	95.9 ± 5.4	0.02
Bowel bother			
Baseline	98.1 ± 3.6	97.7 ± 4.3	0.243
3-month	95.8 ± 7.2	96.7 ± 7.4	0.150
6-month	97.1 ± 6.8	96.9 ± 6.8	0.731
12-month	97.3 ± 4.8	97.2 ± 6.2	0.835
24-month	97.9 ± 4.1	97.5 ± 5.8	0.361
Sexual function			
Baseline	47.8 ± 25.5	53.1 ± 24.7	0.014
3-month	44.5 ± 18.8	22.1 ± 22.5	<0.001
6-month	45.8 ± 20.1	28.9 ± 20.6	<0.001
12-month	46.7 ± 23.4	38.8 ± 22.9	<0.001
24-month	47.5 ± 22.7	49.7 ± 22.3	0.252
Sexual bother			
Baseline	82.1 ± 20.4	80.6 ± 22.8	0.42
3-month	78.7 ± 22.6	62.4 ± 25.7	<0.001
6-month	80.9 ± 21.1	66.3 ± 26.1	<0.001
12-month	81.9 ± 19.5	69.1 ± 23.3	<0.001
24-month	81.3 ± 19.9	77.8 ± 24.8	0.072

Note: EPIC, Expanded Prostate Cancer Index Composite; RP, radical prostatectomy; BT, brachytherapy.

## Data Availability

The data used to support the findings of this study are available from the corresponding author upon request.

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
