# Peer review of "Comparison of the Oncological and Functional Outcomes of Brachytherapy and Radical Prostatectomy for Localized Prostate Cancer"

_medicina, 2022, doi:10.3390/medicina58101387_

Round 1
Reviewer 1 Report
This is a very meaningful manuscript, the study presents that brachytherapy provides equivalent oncological control outcomes in terms of biochemical relapse-free survival and cancer-specific survival for patients with localized prostate cancer compared with radical prostatectomy. The manuscript is well-written and explanatory. Method and Results are well-defined. However, the reviewer suggested that some revision of the manuscript might refine the manuscript.
1, Has this study passed the medical ethics review of the relevant institutions? Please provide the institution and number of the medical ethics approval document.
2, This is not the right way to provide reference of software. Please adopt the following pattern for describing a software: Software or equipment Name, Version (Developer's Name, Town, Country), cat. No. of the reagent. You have made this mistake at every point of the software description.
3, Please update the references, and mainly cite the articles published in the last 5 years.
4, The English language requires further minor examination. Such as, L130 Table 1, High---high; There is a Chinese form of a punctuation mark in the text.
5, Authors should strongly justify the necessity to conduct the described research. This part of the introduction is insufficient. Both, in the introduction and in the discussion, the following paragraphs are often thematically unrelated. Authors should take care of the quality of the text. Lack of description of the statistical methodology employed in the manuscript and needs to be provided.
6, This research was funded by the Institutional Ethical Committee of Clinical Medical College of the Yangzhou University, grant number H201550. The authors should carefully check the details of the fund.
Author Response
Response to Reviewer 1#
This is a very meaningful manuscript, the study presents that brachytherapy provides equivalent oncological control outcomes in terms of biochemical relapse-free survival and cancer-specific survival for patients with localized prostate cancer compared with radical prostatectomy. The manuscript is well-written and explanatory. Method and Results are well-defined. However, the reviewer suggested that some revision of the manuscript might refine the manuscript.
1, Has this study passed the medical ethics review of the relevant institutions? Please provide the institution and number of the medical ethics approval document.
Response: Thanks for your comments. This study was approved by the Medical Ethics Committee of Northern Jiangsu People's Hospital (Approval No. 2016KY-013). Informed consent was obtained from all subjects involved in the study. The medical ethics approval document and the informed consent have been submitted to MS system.
2, This is not the right way to provide reference of software. Please adopt the following pattern for describing a software: Software or equipment Name, Version (Developer's Name, Town, Country), cat. No. of the reagent. You have made this mistake at every point of the software description.
Response: Thanks for your comments. We have revised the text of the MS.
3, Please update the references, and mainly cite the articles published in the last 5 years.
Response: Thanks for your comments. We have updated the refs.
4, The English language requires further minor examination. Such as, L130 Table 1, High---high; There is a Chinese form of a punctuation mark in the text.
Response: Thanks for your comments. We have corrected the errors in the manuscript.
5, Authors should strongly justify the necessity to conduct the described research. This part of the introduction is insufficient. Both, in the introduction and in the discussion, the following paragraphs are often thematically unrelated. Authors should take care of the quality of the text. Lack of description of the statistical methodology employed in the manuscript and needs to be provided.
Response: Thanks for your comments. We have modified the text in the manuscript.
6, This research was funded by the Institutional Ethical Committee of Clinical Medical College of the Yangzhou University, grant number H201550. The authors should carefully check the details of the fund.
Response: Thanks for your comments. We are very sorry; we wrote the ethics approval number of the establishment fund approval number. We have modified the text in the manuscript.
“Funding: This research was funded by the Zhejiang Provincial Basic-Public-Welfare Planning Project, China (grant no. LQ22H230001 to DC), Zhejiang Provincial Medical and Health Sci-Tech Plan Project, China (grant no. 2022KY1256 to FW), Zhejiang Provincial TCM Sci-Tech Project, China (grant no. 2021ZB288 to TS), and Jiangsu Provincial Medical and Health Sci-Tech Plan Project, China (grant no. H201550 to XD). Ethics Approval and Informed Consent Statement: This study was approved by the Medical Ethics Committee of Northern Jiangsu People's Hospital (approval no. 2016KY-013). Informed consent was obtained from all subjects involved in the study.”
Reviewer 2 Report
Congratulations for the authors for this easily readable and good quality article. They aimed to evaluate in a single institutional, retrospective and comparative study oncological and functional outcomes of brachytherapy and radical prostatectomy for localized PCa. They concluded that BT provides equivalent oncological control outcomes in terms of bRFS and CSS for patients with localized PCa compared with RP, and that Gleason ≥8 was the main independent prognostic factor for bRFS. A better HRQoL was found for BT compared to RP, except for urinary and bowel bother.
All in all the manuscript deserves publication. I just have a minor comment:
The limitations part must be developed in the discussion, since many other limitations exist other than the retrospective nature of the study. Examples might be the absence of a propensity score match analysis, the absence of an overall survival analysis as an outcome (I know it is impossible to evaluate on a short-term in PCa, but it is still a limitation), the absence of comparison to numerous focal treatments (ex: HIFU) which are known to also have less side effects and to provide a better quality of life.
Author Response
Response to Reviewer 2#
Congratulations for the authors for this easily readable and good quality article. They aimed to evaluate in a single institutional, retrospective and comparative study oncological and functional outcomes of brachytherapy and radical prostatectomy for localized PCa. They concluded that BT provides equivalent oncological control outcomes in terms of bRFS and CSS for patients with localized PCa compared with RP, and that Gleason ≥8 was the main independent prognostic factor for bRFS. A better HRQoL was found for BT compared to RP, except for urinary and bowel bother.
All in all the manuscript deserves publication. I just have a minor comment:
The limitations part must be developed in the discussion, since many other limitations exist other than the retrospective nature of the study. Examples might be the absence of a propensity score match analysis, the absence of an overall survival analysis as an outcome (I know it is impossible to evaluate on a short-term in PCa, but it is still a limitation), the absence of comparison to numerous focal treatments (ex: HIFU) which are known to also have less side effects and to provide a better quality of life.
Response: Thanks for your comments. According to reviewer’s suggests, the limitation part has been supplemented in the discussion of the manuscript.
Reviewer 3 Report
The authors conducted a retrospective study “Comparison of the oncological and functional outcomes of brachytherapy and radical prostatectomy for localized prostate cancer” and revealed that brachy therapy provides equivalent oncological control outcomes in terms of biochemical relapse-free survival and cancer-specific survival for patients with localized prostate cancer compared with radical prostatectomy. Additionally, Gleason score ≥8 was the main independent prognostic factor for biochemical relapse-free survival. Brachy therapy had better HRQoL compared with radical prostatectomy, except for urinary irritation/obstruction and bowel function or bother, but returned 36 to baseline after 2 years. Following are some advises for revision before possible publication:
1. Statistical analysis part: in line 117, could the authors give explanations why the authors use Mann-Whitney U test to compare medians, and what the medians belong to? The tables did not present median values. Are the data normally distributed or non-normally distributed?
2. Could the authors elaborate on subject inclusion and exclusion criteria in the materials and methods part?
3. The authors showed HRQoL was measured using the Expanded Prostate Cancer Index Composite questionnaire. Could authors elaborate clearly how to evaluate HRQoL in this study?
4. Table 1. Apart from mean scores, the authors should add median values. In addition, could the authors provide the explanation for using median follow-up time instead of mean follow-up time?
5. Line 128-129: Could the authors provide the follow-up time in supplemental material?
6. Reference: The authors should update the reference list during the five latest years.
- Limitation: The study has some limitations. First, this study was only conducted by one institute so it restricts the generalizability. Second, the small number of patients assessed, and the short follow-up period might affect the oncological results and post-treatment health-related quality of life.
Author Response
Response to Reviewer 3#
The authors conducted a retrospective study “Comparison of the oncological and functional outcomes of brachytherapy and radical prostatectomy for localized prostate cancer” and revealed that brachy therapy provides equivalent oncological control outcomes in terms of biochemical relapse-free survival and cancer-specific survival for patients with localized prostate cancer compared with radical prostatectomy. Additionally, Gleason score ≥8 was the main independent prognostic factor for biochemical relapse-free survival. Brachy therapy had better HRQoL compared with radical prostatectomy, except for urinary irritation/obstruction and bowel function or bother, but returned 36 to baseline after 2 years. Following are some advises for revision before possible publication:
- Statistical analysis part: in line 117, could the authors give explanations why the authors use Mann-Whitney U test to compare medians, and what the medians belong to? The tables did not present median values. Are the data normally distributed or non-normally distributed?
Response: Thanks for your comments. According to reviewer’s suggests, we rewrite the description in the Statistical Methods section. “2.5. Statistical analysis Data were expressed as percentage or mean scores ± standard deviation. Differences between categorical variables were compared using the Chi-squared test, and differences between continuous variables were compared using t-test. We used the Kaplan-Meier method and the log-rank test to estimate bRFS and CSS. A Cox regression model was used for multivariate analysis of bRFS. P<0.05 was considered to be statistically significant. All statistical analyses were performed using SPSS Statistics version 23.0 (IBM Corporation, Armonk, NY, USA).”
- Could the authors elaborate on subject inclusion and exclusion criteria in the materials and methods part?
Response: Thanks for your comments. According to reviewer’s suggests, the inclusion and exclusion criteria have been added in the Materials and Methods part. “The inclusion criteria were the following: A clinical T-stage between T1c and T3a, ≥2 years follow‑up post‑treatment. Patients who received adjuvant radiation therapy/chemotherapy and/or patients with distant metastasis were excluded from the present study.”
- The authors showed HRQoL was measured using the Expanded Prostate Cancer Index Composite questionnaire. Could authors elaborate clearly how to evaluate HRQoL in this study?
Response: Thanks for your comments. HRQoL was measured in patients treated for localized PCa with RP and BT using the Expanded Prostate Cancer Index Composite (EPIC) questionnaire at baseline and 3, 6, 12, and 24 months after the treatment. Each domain is scored from 0 to 100, with higher scores indicating better HRQoL.
- Table 1. Apart from mean scores, the authors should add median values. In addition, could the authors provide the explanation for using median follow-up time instead of mean follow-up time?
Response: Thanks for your comments. According to reviewer’s suggests, we have added the median values in Table 1. Furthermore, we used the mean follow-up time. “The mean follow-up time was (58.56±20.59) months (range 24-110 months).”
- Line 128-129: Could the authors provide the follow-up time in supplemental material?
Response: Thanks for your comments. According to reviewer’s suggests, we uploaded the record of the follow-up time as the supplemental material.
- Reference: The authors should update the reference list during the five latest years.
Response: Thanks for your comments. According to reviewer’s suggests, we have updated the references in the manuscript.
- Limitation: The study has some limitations. First, this study was only conducted by one institute so it restricts the generalizability. Second, the small number of patients assessed, and the short follow-up period might affect the oncological results and post-treatment health-related quality of life.
Response: Thanks for your comments. According to reviewer’s suggests, we rewritten the limitation of study. “This study has some limitations. First, this study is a single institutional, retrospective study. The small number of patients evaluated and the short follow-up period may have influenced the oncological results and posttreatment HRQoL. A longer observational period is required for a meaningful comparison of overall survival time. Second, the results of this study may have created some bias due to the lack of propensity score matching analysis. Prospective, randomized studies with a larger number of patients and a longer follow-up period are required to confirm these encouraging results.”